

# Prevalence of and risk factors associated with latent tuberculosis infection in a Latin American region

Javier Andrés Bustamante-Rengifo[1], Luz Ángela González-Salazar[1],
Nicole Osorio-Certuche[1], Yesica Bejarano-Lozano[2],
José Rafael Tovar Cuevas[2], Miryam Astudillo-Hernández[1] and
Maria del Pilar Crespo-Ortiz[1]

[1] Biotechnology and Bacterial Infections Group, Department of Microbiology, Universidad del Valle, Cali, Colombia
[2] Department of Statistics, Universidad del Valle, Cali, Colombia

Corresponding author
Javier Andrés Bustamante-Rengifo,
javier.andres.
bustamante@correounivalle.edu.co

## ABSTRACT

Tuberculosis (TB) represents a health problem in Colombia, and its control is focused on the search for contacts and treatment of TB cases underscoring the role of latent tuberculosis infection (LTBI) as a reservoir of *Mycobacterium tuberculosis*. The burden of LTBI in Colombia is unknown. We aimed to estimate the prevalence of LTBI and identify the associated risk factors. In this cross-sectional study, we recruited participants from four health care centers in Cali, Colombia. The participants were eligible if they were aged between 14 and 70 years, and all participants answered a survey evaluating their medical history and sociodemographic and lifestyle factors. LTBI status was based on tuberculin skin test (TST) positivity using two thresholds: ≥10 mm (TST-10) and ≥15 mm (TST-15). The magnitude of the associations between independent factors and dependent outcomes (LTBI status and TST induration) were evaluated by logistic regression and generalized linear models, respectively. A total of 589 individuals were included with TST positivity rates of 25.3% (TST-10) and 13.2% (TST-15). Logistic regression showed that being between age 40 and 69 years (OR = 7.28, 95% CI [1.62–32.7]), being male (OR = 1.71, 95% CI [1.04–2.84]), being employed (OR = 1.56, 95% CI [1.02–2.38]), and having a low intake of alcohol (OR = 2.40, 95% CI [1.13–5.11]) were risk factors for TST positivity, while living in the north zone (OR = 0.32, 95% CI [0.18–0.55]), living in the suburb zone (OR = 0.28, 95% CI [0.15–0.52]) and having a secondary education (OR = 0.49 95% CI [0.29–0.83]) lowered the risk of TST positivity. The generalized linear model showed that the previous predictors, as well as a low body mass index, had an effect on TST reaction size. The LTBI prevalence found in the population was moderate, reflecting the continuous transmission of *M. tuberculosis*. Social factors seem to play a decisive role in the risk of LTBI. Employed males, who are over 40 years of age, are overweight, have a lower level of education and have a low intake of alcohol (50–100 mL, once/week) should be a priority group for prophylactic treatment as a strategy for TB control in this city.

## INTRODUCTION

Tuberculosis (TB) is an infectious disease caused by the bacillus *Mycobacterium tuberculosis*. In the last 200 years TB has killed more than a billion people, and it is the main cause of death due to a single infectious agent (*Yap et al., 2018*). Antibiotic resistance, lack or poor vaccination campaigns, globalization and poverty have complicated the control of this disease (*Dheda et al., 2014*; *Kargi et al., 2017*). In 2018, the World Health Organization (WHO) reported 10 million new TB cases and 1.2 million TB deaths globally. Eighty percent of TB cases and 70% of deaths occurred in middle-income and low-income countries (*WHO, 2019*). Coevolution with humans has allowed this microorganism to develop mechanisms to persist within the host for decades (*Huynh, Joshi & Brown, 2011*). It is estimated that a quarter of the world's population (~1.7 billion or 23% of people) has latent tuberculosis infection (LTBI) (*Houben & Dodd, 2016*), a state of continuous stimulation of the immune system by *M. tuberculosis* without evidence of clinical symptoms of active disease (*Sharma, Mohanan & Sharma, 2012*; *Yap et al., 2018*). It is thought that 5–10% of these individuals will progress to active TB within the first 2 years, and the risk increases in those patients with suppression of cellular immunity by HIV infection, the use of glucocorticoids, blood or organ transplantation, treatment with tumor necrosis factor α inhibitors, malnutrition or diabetes (*Basera, Ncayiyana & Engel, 2017*; *Chen et al., 2015*; *Huynh, Joshi & Brown, 2011*; *Yap et al., 2018*).

The WHO South-East Asia, Western Pacific and Africa regions have an LTBI prevalence above 20%, while the Eastern Mediterranean, Europe and The Americas have an LTBI prevalence lower than 17%. The Americas region has the lowest prevalence of LTBI, with 11% or approximately 108,000 million infected people (*Houben & Dodd, 2016*). However, in the Americas, TB persists as a major public health problem, approximately 282,000 new TB cases are reported every year, and approximately 18,000 people die from this cause (*WHO, 2018b*). Countries showing the highest morbidity and mortality are Brazil, Peru, Mexico, Haiti and Colombia, representing 68% of the TB cases in the region (*PAHO, 2018*).

Colombia is the third most populated country in Latin America with a population of 49 million inhabitants. Despite the country's efforts to control TB, its incidence has increased in the last decade, from 26 cases per 100,000 inhabitants in 2008 (*Rodríguez, Gil & Vera, 2010*) to 33 cases per 100,000 in 2018 (*WHO, 2018a*). Local incidence rates highly differ between regions. Cali, is one of the cities with the highest TB incidence (41 cases per 100,000 inhabitants) which is higher than the national average (26.5 cases per 100,000 inhabitants) (*Lesmes Duque & Reina, 2016*; *Sivigila, 2017*), and Cali has an annual risk of tuberculosis infection of 1.3%, which is higher than the 0.1% risk reported for the world population (*De la Pava, Salguero & Alzate, 2002*; *Hassan & Diab, 2014*). Two contributing factors for this particular setting are (a) continued community transmission due to the delay in diagnosing cases of active TB, and (b) reactivation of LTBI in vulnerable populations (household TB contacts, prisoners, health care workers, immunocompromised patients, children under 5 years old and the elderly) (*Barbosa et al., 2014*; *Del Corral et al., 2009*).

Latent tuberculosis infection seems to be a reservoir from which active TB will emerge (*Abubakar et al., 2018*), representing a challenge to the aims of the End TB Strategy (90% reduction in TB incidence by 2035) (*Houben & Dodd, 2016*). However, detecting LTBI in humans is difficult. Historically, LTBI detection has been performed by assessing the T-cell response against *M. tuberculosis* using the tuberculin skin test (TST) or interferon-gamma release assay (IGRA). These tests are useful to identify people who could benefit from prophylaxis and to control TB incidence and transmission (*Abubakar et al., 2018*). Nevertheless, both the TST and IGRA have sensitivity and specificity limitations (*Muñoz, Stagg & Abubakar, 2015*), and a challenge to understanding how to best use these test is the lack of a gold standard (*Stout et al., 2018*). Although the evidence shows that both the TST and IGRA can be used for LTBI detection, the availability and affordability of the tests will determine which one should be chosen (*WHO, 2018c*). In Colombia, the test of choice is the TST, as it requires fewer resources than the IGRA and is more familiar to professionals in clinical settings (*Ministry of Health of Colombia, 2015*).

The TST interpretation should consider the presence of immunosuppressive conditions and TB prevalence in the setting, so different cut-off values have been recommended for defining a positive TST: ≥5 mm in immunosuppressed patients and close contact of TB cases and ≥10 mm in healthy individuals living in high-prevalence areas (*ATS, 2000*; *Muñoz, Stagg & Abubakar, 2015*). Considering a possible cross-reactivity with Bacillus Calmette-Guérin (BCG) vaccination and non-tuberculosis mycobacterial infections, some authors suggest using a threshold of 15 mm or more (*Abubakar et al., 2018*; *Shero et al., 2014*). In fact, TST reactivity stratified by BCG vaccination status have shown a positive predictive value for the development of TB similar to IGRA (*Abubakar et al., 2018*), and large TST reactions are more likely to indicate a higher risk of developing active disease (*Cao et al., 2019*; *Moran-Mendoza et al., 2007*; *Watkins, Brennan & Plant, 2000*). All these considerations must be taken into account when comparing the TST properties between different studies.

There are limited data on the epidemiology and risk of LTBI in the general population, particularly in developing countries. Additionally, LTBI screening is only recommended in target groups and extensive testing of LTBI status in the general population is not affordable (*Chen et al., 2015*). In Colombia, there is only one published study that describes TST reactivity in the source population (*Del Corral et al., 2009*). For this reason, the aim of this study was to estimate the prevalence of LTBI using two different TST thresholds (≥10 mm and ≥15 mm) and to identify the risk factors associated with a positive TST result and TST induration size. This may allow us to determine the potential size of the current reservoir of infection and to provide information to implement control measures.

## MATERIALS AND METHODS

### Study design

This cross-sectional study was conducted from September 27, 2016 to December 1, 2017 in Cali, Colombia. Cali is the third most important city in the country with a population of 2,394,925 inhabitants distributed in 22 districts. The estimated TB incidence is 41 cases per 100,000 inhabitants (*Lesmes Duque & Reina, 2016*). The city's public health system is

divided into five government social companies (the acronym in Spanish is ESE), and each company provides primary health-care services to different districts by zones, as follows: ESE center (districts 8, 9, 10, 11 and 12), ESE suburb (districts 1, 3, 17, 18, 19, 20 and 22), ESE north (districts 2, 4, 5, 6 and 7), ESE east (districts 13, 14, 15 and 21) and ESE southeast (district 16). For this study, as a strategy to enroll participants representative of the general population, three reference primary health-care facilities were selected, one for each ESE except for the ESE east and ESE southeast, which were not included for safety and accessibility reasons. To achieve greater representativeness of the center zone, a secondary-care facility was also included. The selected facilities provide health services to approximately 50% of the city's low-income population. Healthy individuals attending general medical examinations or the dental service, along their companions, were invited to participate in the research.

## Participants and data collection

A total of 1,079 volunteers with ages ranging from 14 to 70 years completed a standardized questionnaire including demographic, socioeconomic and clinical data and lifestyle habits. Considering an estimated 42.7% LTBI (*Del Corral et al., 2009*) with a 95% confidence level, 4% precision and 10% nonresponse rate, the sample size calculated was 647 individuals. Exclusion criteria for this study were a history of chronic diseases (hypertension, diabetes and cancer), previous TB disease or chronic respiratory symptoms, heart disease, advanced liver disease and immunosuppressive conditions (HIV/AIDS infection, transplant and lupus), pregnancy or lactation. These conditions were ruled out to reduce anergy to TST and the risk of severe arm swelling after TST application (*Muñoz, Stagg & Abubakar, 2015*). Thus, 629 (58.3%) subjects were subjected to a TST. However, 589 individuals were included for further study (Fig. 1).

Overcrowding was defined as homes with more than three to less than five people per room. BCG vaccination status was evaluated by visual inspection (vaccination scar). The household socioeconomic status (SES) was defined as a composite index developed by an analysis based on characteristics of the dwelling, source of drinking water, type of toilet facilities and features of the neighborhood. The household SES index was categorized into tertiles of (1) extremely-low, (2) low and (3) medium, as this was a population with low resources. A history of smoking was defined as tobacco use in the last 6 months. Alcohol consumption was stratified into four levels: high (consumption between 350 and 750 mL, ≥4 times/week), moderate (consumption between 250 and 750 mL, 2–3 times/week), low (consumption between 50 and 100 mL, once/week) and none. Physical activity was defined as yes (exercise every day or ≥3 times/week) or no (never exercise or <2 times/week). The body-mass index (BMI) was classified as underweight (<18.5 kg/m$^2$), normal weight (≥18.5 to 24.9 kg/m$^2$), overweight (≥25 to 29.9 kg/m$^2$) and obesity (≥30 kg/m$^2$) according to the classification proposed by the WHO (*WHO, 2003*). The employment situation was categorized as employe (self-employed workers and subordinate workers) and unemployed (housewives, student or retired).

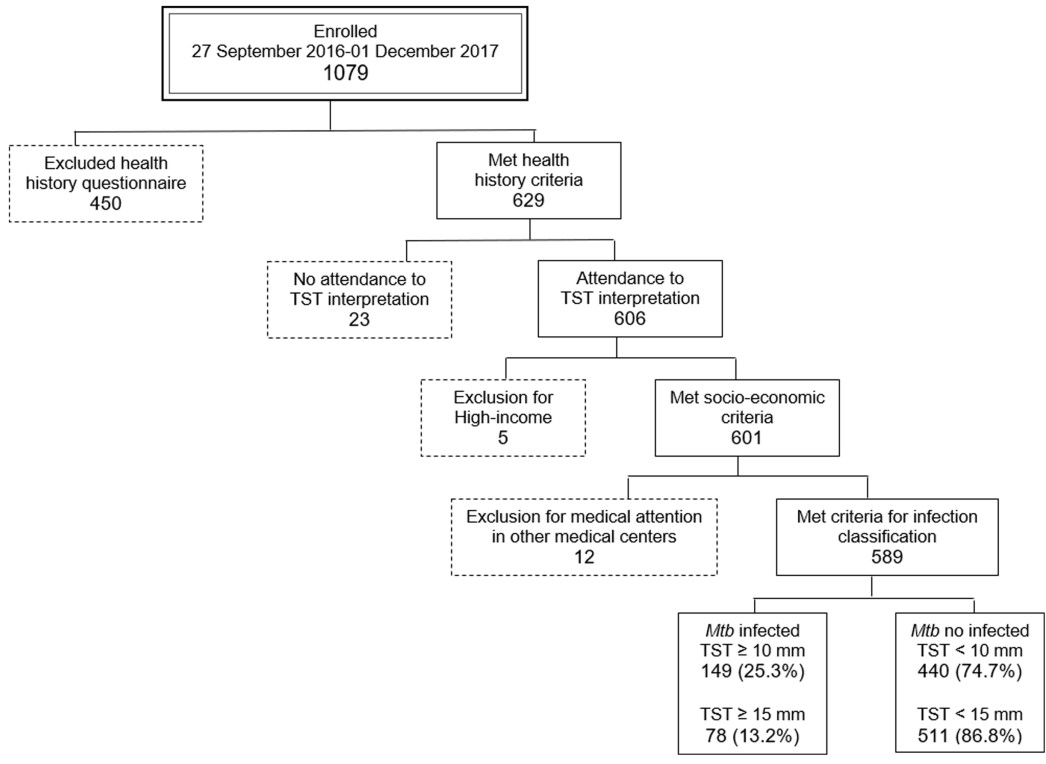

**Figure 1 Flow chart for the enrollment and LTBI diagnosis on the study population.**

## Ethics statement

The study protocol was approved by the ethics committee of the Universidad del Valle-CIREH (#008-015) (Cali, Colombia). All participants provided written informed consent.

## Tuberculin skin test

Five units of tuberculin purified protein derivate (PPD) of *M. tuberculosis* Mammalian® (BB-NCIPD Ltd., Sofia, Bulgaria) in 0.1 mL was injected into in the dorsal surface of the forearm. The induration was measured 48–72 h after injection (*Rieder et al., 2011*).

## Determination of latent infection by *M. tuberculosis*

Two different thresholds were considered for a positive TST result. An induration of 10 mm or more (TST-10) and a BCG-dependent induration of 15 mm or more (TST-15; for the BCG vaccinated participants) were positive according to the current CDC guidelines (*ATS, 2000*). Given that vaccination actions in Colombia began in the 1960s, regular administration of the BCG vaccine was intensified in the 1970s as part of public health strategies against preventable diseases defined by the PAHO/WHO (*Ministry of Health, 2000*). For the primary analysis, if the BCG status was unknown, it was assumed that the individuals had been vaccinated (consistent with international recommendations) (*Abubakar et al., 2018*). All the participants with a positive TST result had a standard anteroposterior and lateral chest radiograph (CXR). LTBI was defined as a positive TST

result in the absence of TB respiratory symptoms and normal radiological findings. Active TB was suspected if the individuals had radiological abnormalities, chronic cough (more than 3 weeks), weight loss, night sweats, and fever. All cases with suspected TB were excluded and referred to the tuberculosis program for TB treatment.

## Statistical analysis

Kolmogorov–Smirnov's test revealed that the quantitative data did not follow a normal distribution, so nonparametric statistics were used. Continuous data were compared using the Mann–Whitney $U$ test or Kruskal–Wallis test. Dunn's test was used to analyze variables with more than two categories, and the significance values were adjusted by Bonferroni correction. To analyze the association between LTBI status and the independent variables, Pearson's Chi-square test or Fisher's exact test with odds ratio (OR) and 95% confidence intervals were used. The trend analysis for TST positivity with increasing age for both males and females was performed using a chi-square test for trend.

Multivariate logistic regression or generalized linear regression models were used to determine the associations between the independent variables and TST positivity or TST induration, respectively. In the logistic regression model, the backward Wald method was used, controlling each factor. The selection of predictors within the model was performed using the likelihood criteria (input $p \leq 0.05$, output $p \geq 0.10$). In the generalized linear model, a custom model with a Poisson distribution and a Log link function was used. The model was constructed using the main effects method, and parameter estimates to select the best model were calculated using the Pearson chi-square method accompanied by a robust covariance estimator. The β coefficients reported were standardized. For all models, bootstrap analysis was performed with 5,000 samples to compare the effect measures obtained in the original model with the bootstrapped model. The analyses were performed using the statistical package SPSS 24.0 (SPSS Inc., Chicago, IL, USA).

## RESULTS

### Sociodemographic characteristics and lifestyle habits

A total of 589 participants were included in this study for a response rate of 54.6% (Fig. 1). As seen in Table 1, age was classified into six groups from 14 to 70 years old. Fifty-five percent of the individuals were in the age range of 40–59 years, and the average age was 43.8 ± 13.2 years. The females constituted 80% of the total population. The distribution of the individuals within the three household SES categories was homogeneous (~30%). A total of 5.4% of participants lived in overcrowded areas. More than half of the individuals had completed high school studies, were unemployed, and lived in the northern zone. In relation to the lifestyle of this population, the majority were overweight or obese, did not perform regular physical activity, had no history of smoking or current cigaret consumption, and did not consume alcohol. Only 9% of the individuals had no visible BCG scar.

### Prevalence of latent tuberculosis infection

The overall prevalence of latent tuberculosis using a threshold ≥10 mm (TST-10) was 25.3% (149/589; 95% CI [21.7–28.3%]), and with a threshold ≥15 mm (TST-15) was 13.2%

**Table 1 Prevalence of LTBI according to sociodemographic and lifestyle habits in the study population and its association with TST induration size (n = 589).**

| Characteristics | n (%) | TST-10 (%) | TST-15 (%) | TST (millimeters) | | |
|---|---|---|---|---|---|---|
| | | | | Mean | SD | p |
| Age (years) | | | | | | |
| Mean (SD) | 43.8 (13.2) | | | | | |
| 14–19 | 31 (5.3) | 6.5 | 6.5 | 3.42 | 4.20 | **0.000** |
| 20–29 | 74 (12.6) | 13.5 | 5.4 | 4.15 | 4.76 | |
| 30–39 | 96 (16.3) | 22.9 | 11.5 | 5.32 | 5.63 | |
| 40–49 | 145 (24.6) | 31.0 | 13.8 | 6.95 | 6.73 | |
| 50–59 | 180 (30.6) | 26.7 | 16.7 | 6.87 | 7.01 | |
| 60–69 | 63 (10.7) | 34.9 | 17.5 | 8.05 | 7.82 | |
| Sex | | | | | | |
| Male | 118 (20.0) | 33.9 | 17.8 | 7.03 | 6.81 | 0.267 |
| Female | 471 (80.0) | 23.1 | 12.1 | 6.04 | 6.49 | |
| Household SES | | | | | | |
| 1 | 212 (36.0) | 22.6 | 9.4 | 5.83 | 5.59 | 0.251 |
| 2 | 198 (33.6) | 23.2 | 15.2 | 5.97 | 6.63 | |
| 3 | 179 (30.4) | 30.7 | 15.6 | 7.03 | 7.46 | |
| Overcrowding | | | | | | |
| >3 person/room | 32 (5.4) | 25.0 | 15.6 | 6.06 | 6.04 | 0.766 |
| ≤3 person/room | 557 (94.6) | 25.3 | 13.1 | 6.25 | 6.60 | |
| Educational status | | | | | | |
| Primary or less | 215 (36.5) | 32.1 | 18.6 | 7.58 | 7.32 | **0.001** |
| Secondary | 321 (54.5) | 21.5 | 10.6 | 5.52 | 6.07 | |
| Postsecondary | 53 (9.0) | 20.8 | 7.5 | 5.15 | 5.25 | |
| Employment situation | | | | | | |
| Employee | 259 (44.0) | 32.0 | 16.2 | 7.08 | 7.21 | **0.034** |
| Unemployed | 330 (56.0) | 20.0 | 10.9 | 5.58 | 5.93 | |
| Zone | | | | | | |
| North | 334 (56.7) | 22.2 | 10.8 | 5.63 | 6.13 | **0.001** |
| Suburb | 183 (31.1) | 21.3 | 9.3 | 5.71 | 5.37 | |
| Central | 72 (12.2) | 50 | 34.7 | 10.44 | 9.26 | |
| Physical activity | | | | | | |
| No | 412 (69.9) | 23.1 | 12.6 | 5.95 | 6.27 | 0.414 |
| Yes | 177 (30.1) | 30.5 | 22.2 | 6.92 | 7.17 | |
| Ever smoked | | | | | | |
| Yes | 89 (15.1) | 29.2 | 14.6 | 6.90 | 6.95 | 0.311 |
| No | 500 (84.9) | 24.6 | 13.0 | 6.12 | 6.50 | |
| Current smoking | | | | | | |
| Yes | 68 (11.5) | 29.4 | 16.2 | 7.00 | 7.37 | 0.641 |
| No | 521 (88.5) | 24.8 | 12.9 | 6.14 | 6.45 | |
| Alcohol consumption | | | | | | |

(Continued)

| Table 1 (continued) | | | | | | |
| --- | --- | --- | --- | --- | --- | --- |
| Characteristics | n (%) | TST-10 (%) | TST-15 (%) | TST (millimeters) | | |
| | | | | Mean | SD | p |
| High | 6 (1.0) | 16.7 | 0.0 | 5.33 | 4.23 | **0.012** |
| Moderate | 35 (5.9) | 17.1 | 8.6 | 4.46 | 5.77 | |
| Low | 35 (5.9) | 42.8 | 25.7 | 6.46 | 8.47 | |
| None | 513 (87.1) | 24.8 | 12.9 | 6.16 | 6.44 | |
| BMI (kg/m²) | | | | | | |
| <18.5 | 20 (3.4) | 5.0 | 5.0 | 3.35 | 3.63 | **0.025** |
| ≥18.5–24.9 | 248 (42.1) | 22.6 | 12.9 | 5.81 | 6.57 | |
| ≥25–29.9 | 229 (38.9) | 28.4 | 14.0 | 6.70 | 6.53 | |
| ≥30 | 92 (15.6) | 29.3 | 14.1 | 6.89 | 6.94 | |
| BCG scar | | | | | | |
| Yes | 536 (91.0) | 25.4 | 15.1 | 6.25 | 6.52 | 0.670 |
| No | 53 (9.0) | 24.5 | 13.1 | 6.13 | 7.03 | |

**Notes:**
BCG, Bacillus Calmette-Guérin; BMI, body mass index; TST, Tuberculin Skin Test; CI, confidence interval; OR, odds ratio; SD, Standard deviation.
Significant variables are denoted in bold.

(78/589; 95% CI [10.4–15.5%]). There was a gradual increase in the LTBI prevalence with age, reaching the highest prevalence in the 60–69 years category. The LTBI prevalence was higher in males (33.9% for TST-10 and 17.8% for TST-15) (Table 1).

The LTBI prevalence determined by TST-10 and TST-15 was higher in individuals with medium household SES, individual with a primary education level, employes, residents of the central zone, smokers, light drinkers, individual who perform regular physical activity, and overweight and obese individuals. In contrast, no differences were observed in the LTBI prevalence with respect to overcrowding within households or BCG status (Table 1).

## Demographic, socioeconomic, and lifestyle factors and their association with TST induration

The average induration size was signification different between TST+ and TST− individual when using either the ≥10 mm (16 mm vs 2.9 mm) or ≥15 mm (19.7 mm vs 4.2 mm) threshold ($p = 0.000$ for both), and the mean was located above the cut-off point established for each threshold. Independent of threshold, the TST induration size showed an association with multiple variables including age, educational status, employment situation, residence zone, alcohol consumption and BMI (Table 1).

The average induration in individuals under 29 years old was significantly different from that of individuals over 30 years old. Individuals with a primary education level had a higher average TST induration size than individuals with higher education ($p < 0.05$). It was also observed that individuals residing in the center of city presented a higher average in TST induration size compared to individuals residing in the northern ($p = 0.000$) and suburb ($p = 0.040$) zones. Individuals with low alcohol consumption had a

**Table 2 Generalized linear model on risk factors for TST induration size.**

| Variables | *β | SD β | 95% CI | p | B β | B SD β | B 95% CI | B p |
|---|---|---|---|---|---|---|---|---|
| Intercept | 1.965 | 0.277 | [1.42–2.51] | **0.000** | 1.965 | 0.293 | [1.35–2.49] | **0.000** |
| Education (secondary) | −0.216 | 0.097 | [−0.40 to −0.02] | **0.026** | −0.216 | 0.100 | [−0.41 to −0.02] | **0.034** |
| Education (postsecondary) | −0.312 | 0.151 | [−0.61 to −0.01] | **0.039** | −0.312 | 0.160 | [−0.66 to −0.00] | **0.054** |
| BMI (<18.5 kg/m²) | −0.574 | 0.194 | [−0.95 to −0.19] | **0.003** | −0.574 | 0.215 | [−1.04 to −0.22] | **0.015** |
| Zone (north) | −0.472 | 0.129 | [−0.72 to −0.22] | **0.000** | −0.472 | 0.135 | [−0.73 to −0.20] | **0.001** |
| Zone (suburb) | −0.499 | 1.137 | [−0.77 to −0.23] | **0.000** | −0.499 | 0.143 | [−0.78 to −0.22] | **0.001** |
| Alcohol consumption (low) | 0.405 | 0.164 | [0.08– 0.73] | **0.014** | 0.405 | 0.175 | [0.03–0.73] | **0.029** |
| Age (30–39 years) | 0.338 | 0.241 | [−0.13 to 0.81] | 0.162 | 0.338 | 0.248 | [−0.12 to 0.86] | 0.193 |
| Age (40–49 years) | 0.561 | 0.235 | [−0.10 to 1.02] | **0.017** | 0.561 | 0.245 | [0.08–1.07] | **0.050** |
| Age (50–59 years) | 0.421 | 0.236 | [−0.04 to 0.89] | 0.075 | 0.421 | 0.247 | [−0.03 to −0.93] | 0.122 |
| Age (60–69 years) | 0.666 | 0.255 | [0.17–1.17] | **0.009** | 0.666 | 0.268 | [0.17–1.20] | **0.003** |

Notes:
* Standardized Coefficients, SD: Standard deviation; B: Bootstrapped for 5,000 samples.
* Significant variables are denoted in bold.

greater TST induration size compared to individuals with moderate consumption ($p = 0.006$). Regarding BMI, it was initially observed that individuals who were underweight had a lower TST induration that overweight and obese individuals ($p < 0.05$). However, after adjustment of the significance values no statistical differences were observed.

In the generalized linear model, age above 40 years old, residing in the northern or suburb zone, having higher education (secondary and postsecondary), low alcohol consumption and lower BMI showed an association with TST induration. The coefficient of determination found was 0.161 (16.1%). The bootstrapping results confirmed the estimates obtained with the generalized linear regression model. Table 2 shows the standardized β coefficients and likelihood values.

## Demographic, socioeconomic, lifestyle factors and their association with TST positivity (≥10 mm)

In the bivariate analysis present in Table 3, when the 14–19 years age category was taken as the reference group, it was found that after the age of 30 years, the risk of LTBI increased significantly, with ORs from 4.31 to 7.78. In this cross-sectional study, males had a 0.70-fold higher risk of LTBI than females (OR = 1.70, 95% CI [1.10–2.64]). Meanwhile, as shown in Fig. 2, TST-10 positivity increased in males and females with increasing age, a trend that was significant ($p$ for trend <0.05). However, the LTBI prevalence in males was significantly higher than in females for the 40–49 years age category (OR = 3.72, 95% CI [1.38–10.03]) TST-10 (Table 4). The education distribution showed that a higher education level decreased the risk of LTBI when compared with a primary or lower level (OR = 0.59, 95% CI [0.39–0.86] for secondary). Being an employe significantly increased the LTBI risk (OR = 1.89, 95% CI [1.30–2.75]). When the risk of LTBI by zone was evaluated with reference to the high prevalence center zone (50%), it was observed that living in the north or suburb zone decreased the LTBI risk (OR = 0.29, 95% CI [0.17–0.48] and

Table 3 Characteristics of the study population and its association with TST-10 and TST-15 positivity.

| Characteristics | n | TST-10 | | | TST-15 | | |
|---|---|---|---|---|---|---|---|
| | | n (%) | OR [95% CI] | p | n (%) | OR [95%CI] | p |
| Age (years) | | | | | | | |
| 14–19 | 31 | 2 (1.3) | Reference | | 2 (2.6) | Reference | |
| 20–29 | 74 | 10 (6.7) | 2.26 [0.46–11.0] | 0.300 | 4 (5.1) | 0.83 [0.14–4.77] | 1.00 |
| 30–39 | 96 | 22 (14.8) | 4.31 [0.95–19.5] | 0.042 | 11 (14.1) | 1.87 [0.39–8.97] | 0.424 |
| 40–49 | 145 | 45 (30.2) | 6.52 [1.49–28.5] | 0.005 | 20 (25.6) | 2.32 [0.51–10.5] | 0.262 |
| 50–59 | 180 | 48 (30.2) | 5.27 [1.21–22.9] | 0.014 | 30 (38.5) | 2.90 [0.65–12.8] | 0.143 |
| 60–69 | 63 | 22 (14.8) | 7.78 [1.69–35.7] | 0.003 | 11 (14.1) | 3.06 [0.63–14.7] | 0.146 |
| Sex | | | | | | | |
| Male | 118 | 40 (26.8) | 1.70 [1.10–2.64] | 0.016 | 21 (26.9) | 1.57 [0.91–2.72] | 0.103 |
| Female | 471 | 109 (73.2) | Reference | | 57 (73.1) | Reference | |
| Household SES | | | | | | | |
| 1 | 212 | 48 (32.2) | 0.66 [0.42–1.04] | 0.071 | 20 (25.6) | 0.56 [0.30–1.03] | 0.062 |
| 2 | 198 | 46 (30.9) | 0.68 [0.43–1.08] | 0.101 | 30 (38.5) | 0.96 [0.55–1.68] | 0.895 |
| 3 | 179 | 55 (36.9) | Reference | | 28 (35.9) | Reference | |
| Overcrowding | | | | | | | |
| >3 person/room | 32 | 8 (5.4) | 0.98 [0.43–2.24] | 0.968 | 5 (6.4) | 1.23 [0.45–3.29] | 0.683 |
| ≤3 person/room | 557 | 141 (94.6) | Reference | | 73 (93.6) | Reference | |
| Educational status | | | | | | | |
| Postsecondary | 53 | 11 (7.4) | 0.55 [0.27–1.14] | 0.106 | 4 (5.1) | 0.35 [0.12–1.05] | 0.052 |
| Secondary | 321 | 69 (46.3) | 0.59 [0.39–0.86] | 0.006 | 34 (43.6) | 0.52 [0.32–0.85] | 0.008 |
| Primary or less | 215 | 69 (46.3) | Reference | | 40 (51.3) | Reference | |
| Employment situation | | | | | | | |
| Employee | 259 | 83 (55.7) | 1.89 [1.30–2.75] | 0.001 | 42 (53.8) | 1.58 [0.98–2.55] | 0.059 |
| Unemployed | 330 | 66 (44.3) | Reference | | 36 (46.2) | Reference | |
| Zone | | | | | | | |
| North | 334 | 74 (49.7) | 0.29 [0.17–0.48] | 0.000 | 36 (46.2) | 0.22 [0.12–0.41] | 0.000 |
| Suburb | 183 | 39 (26.2) | 0.27 [0.15–0.49] | 0.000 | 17 (21.8) | 0.19 [0.09–0.38] | 0.000 |
| Central | 72 | 36 (24.2) | Reference | | 25 (32.1) | Reference | |
| Physical activity | | | | | | | |
| No | 412 | 95 (63.8) | 0.68 [0.46–1.01] | 0.057 | 52 (66.7) | 0.83 [0.50–1.39] | 0.497 |
| Yes | 177 | 541 (36.2) | Reference | | 26 (33.3) | Reference | |
| Ever smoked | | | | | | | |
| Yes | 89 | 26 (17.4) | 1.27 [0.77–2.09] | 0.356 | 13 (16.7) | 1.14 [0.60–2.18] | 0.680 |
| No | 500 | 123 (82.6) | Reference | | 65 (83.3) | Reference | |
| Current smoking | | | | | | | |
| Yes | 68 | 20 (13.4) | 1.27 [0.72–2.21] | 0.407 | 11 (14.1) | 1.31 [0.65–2.62] | 0.448 |
| No | 521 | 129 (86.6) | Reference | | 67 (85.9) | Reference | |
| Alcohol consumption | | | | | | | |
| High | 6 | 1 (0.7) | 0.61 [0.07–5.25] | 0.648 | 0 (0.0) | – | 0.347 |
| Moderate | 36 | 6 (4.0) | 0.61 [0.25–1.49] | 0.273 | 3 (3.8) | 0.62 [018–2.06] | 0.428 |

| Table 3 (continued) | | | | | | | | |
|---|---|---|---|---|---|---|---|---|
| Characteristics | n | TST-10 | | | | TST-15 | | |
| | | n (%) | OR [95% CI] | p | | n (%) | OR [95%CI] | p |
| Low | 35 | 15 (10.1) | 2.28 [1.13–4.59] | **0.018** | | 9 (11.5) | 2.34 [1.05–5.22] | **0.032** |
| None | 513 | 127 (85.2) | Reference | | | 66 (84.6) | Reference | |
| BMI (kg/m$^2$) | | | | | | | | |
| <18.5 | 20 | 1 (0.7) | 0.18 [0.02–1.38] | 0.065 | | 1 (1.3) | 0.35 [0.04–2.74] | 0.301 |
| ≥18.5–24.9 | 248 | 56 (37.6) | Reference | | | 32 (41.0) | Reference | |
| ≥25–29.9 | 229 | 65 (43.6) | 1.35 [0.89–2.05] | 0.146 | | 32 (41.0) | 1.09 [0.65–1.86] | 0.732 |
| ≥30 | 92 | 27 (18.1) | 1.42 [0.83–2.44] | 0.197 | | 13 (16.7) | 1.11 [0.55–2.22] | 0.767 |
| BCG Scar | | | | | | | | |
| No | 53 | 13 (8.7) | 0.96 [0.49–1.84] | 0.893 | | 8 (10.3) | 1.18 [0.53–2.61] | 0.677 |
| Yes | 536 | 136 (91.3) | Reference | | | 70 (89.7) | Reference | |

Note:
Significant variables are denoted in bold

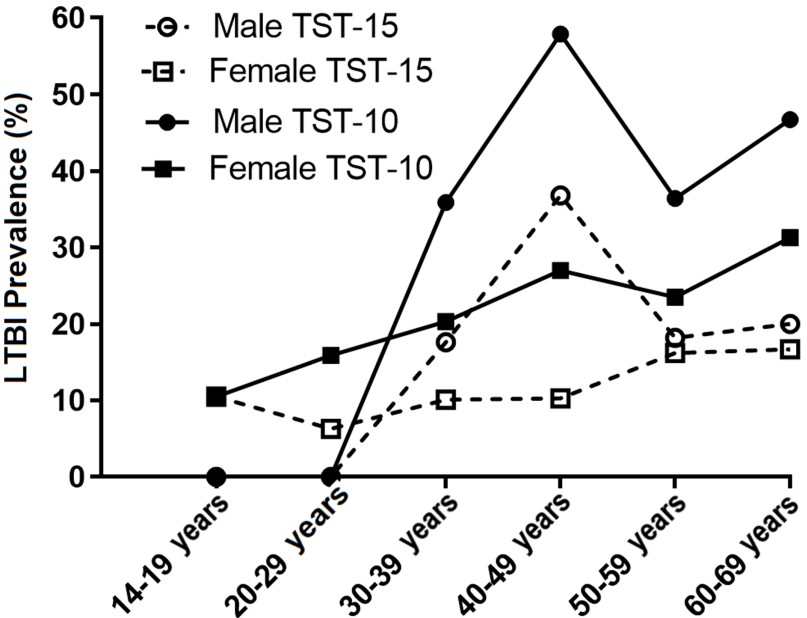

**Figure 2 Age-specific trend of LTBI in the study population stratified by sex.**

OR= 0.27, 95% CI [0.15–0.49], respectively). Low alcohol consumption significantly increased the risk of LTBI (OR = 2.28, 95% CI [1.13–4.59]) (Table 3). No association was observed between BCG scar and TST positivity.

In the multivariate logistic regression analysis (Table 5), being age 30 years and above (ORs from 4.29 to 8.30), being male (OR = 1.71, 95% CI [1.04–2.84]), being employed (OR = 1.56, 95% CI [1.02–2.38]) and having a low alcohol consumption (OR = 2.40, 95% CI [1.13–5.11]) were risk factors for LTBI, while living in the north or suburb zone of the city reduced LTBI risk (OR = 0.32, 95% CI [0.18–0.55] and OR = 0.28, 95% CI [0.15–0.52], respectively). The model had 12.8% sensitivity and 97.3% specificity with a

**Table 4 LTB prevalence between males and females by age groups using a TST-10 threshold.**

| Age (years) | Male | | | | Female | | | | OR (95% CI) | p |
|---|---|---|---|---|---|---|---|---|---|---|
| | n | TST-positive | | p | n | TST-positive | | p | | |
| | | n | (%) | | | n | (%) | | | |
| 14–19 | 12 | 0 | 0 | | 19 | 2 | 10.5 | | – | 0.510* |
| 20–29 | 11 | 0 | 0 | | 63 | 10 | 15.9 | | – | 0.155 |
| 30–39 | 17 | 6 | 35.3 | **0.001**‡ | 79 | 16 | 20.3 | **0.026**‡ | 2.15 [0.69–6.69] | 0.181 |
| 40–49 | 19 | 11 | 57.9 | | 126 | 34 | 27.0 | | 3.72 [1.38–10.0] | **0.007** |
| 50–59 | 44 | 16 | 36.4 | | 136 | 32 | 23.5 | | 1.86 [0.89–3.86] | 0.094 |
| 60–69 | 15 | 7 | 46.7 | | 48 | 15 | 31.3 | | 1.93 [0.59–6.23] | 0.274 |

Notes:
* $p$ value was calculated from Fisher's exact test.
‡ $p$ value for trend was calculated from Chi-square test.
Significant variables are denoted in bold.

**Table 5 Risk factors associated with TST-10 positivity using a logistic regression.**

| Variables | β | Wald | OR* | 95% CI | p | B p |
|---|---|---|---|---|---|---|
| Intercept | −2.007 | 6.670 | | | **0.010** | **0.007** |
| Sex (male) | 0.540 | 4.420 | 1.71 | [1.04–2.84] | **0.036** | **0.027** |
| Employment situation (employee) | 0.445 | 4.272 | 1.56 | [1.02–2.38] | **0.039** | **0.040** |
| Zone (north) | −1.153 | 16.185 | 0.32 | [0.18–0.55] | **0.000** | **0.000** |
| Zone (suburb) | −1.267 | 16.547 | 0.28 | [0.15–0.52] | **0.000** | **0.000** |
| Alcohol consumption (low) | 0.874 | 5.139 | 2.40 | [1.13–5.11] | **0.023** | **0.032** |
| Age (30–39 years) | 1.455 | 3.414 | 4.29 | [0.92–20.0] | 0.065 | **0.034** |
| Age (40–49 years) | 1.986 | 6.719 | 7.28 | [1.62–32.7] | **0.010** | **0.008** |
| Age (50–59 years) | 1.514 | 3.929 | 4.55 | [1.02–20.3] | **0.047** | **0.025** |
| Age (60–69 years) | 2.116 | 7.160 | 8.30 | [1.76–39.1] | **0.007** | **0.004** |

Notes:
* From a multivariate logistic regression model with age, sex, social level, overcrowding, educational status, employment situation, zone, exercise, ever smoking, current smoking, alcohol consumption, BMI and BCG scar.
B: Bootstrapped for 5,000 samples.
Significant variables are denoted in bold.

Cox & Snell $R$-square coefficient of 0.098 (9.8%). The bootstrapping results confirmed the estimates obtained with the initial model and corroborated the significance of the predictor (age 30–39 years) within the model.

## Demographic, socioeconomic, and lifestyle factors and their association with TST positivity (≥15 mm)

In the bivariate analysis for the TST-15 threshold, only three variables showed an association with TST positivity (Table 3). It was observed that a higher education level (secondary) (OR = 0.52, 95% CI [0.32–0.85]) and living in the north zone (OR = 0.22, 95% CI [0.12–0.41]) or suburb zone (OR = 0.19, 95% CI [0.09–0.38]) reduced the risk of LTBI. In contrast, low alcohol consumption was a risk factor for LTBI (OR = 2.34, 95% CI [1.05–5.22]). Although a significant association with TST-15 positivity was not found for any of the age categories or sex, Fig. 2 shows an increasing trend in the LTBI prevalence

**Table 6 Risk factors associated with TST-15 positivity using a logistic regression.**

| Variables | β | Wald | OR* | 95% CI | p | B p |
|---|---|---|---|---|---|---|
| Intercept | −0.421 | 2.011 | | | 0.156 | 0.168 |
| Sex (male) | 0.564 | 3.534 | 1.76 | [0.98–3.17] | 0.060 | 0.085 |
| Education (secondary) | −0.711 | 7.066 | 0.49 | [0.29–0.83] | **0.008** | **0.008** |
| Zone (north) | −1.381 | 19.241 | 0.25 | [0.14–0.47] | **0.000** | **0.000** |
| Zone (suburb) | −1.653 | 20.31 | 0.19 | [0.09–0.39] | **0.000** | **0.000** |
| Alcohol consumption (low) | 0.845 | 3.619 | 2.33 | [0.98–5.56] | 0.057 | 0.080 |

Notes:
* From a multivariate logistic regression model with age, sex, social level, overcrowding, educational status, employment situation, zone, exercise, ever smoking, current smoking, alcohol consumption, BMI and BCG scar.
B: Bootstrapped for 5,000 samples.
Significant variables are denoted in bold.

with age that was significant in females ($p = 0.036$) but not in males ($p = 0.077$). However, a positive TST result was more likely in males than in females for the 40–49 years age category (OR = 5.07, 95% CI [1.69–15.1]) (Table S1).

In the multivariate analysis (Table 6), in agreement with the regression model obtained for TST-10, it was observed that being male (OR = 1.76, 95% CI [0.98–3.17]) and having a low alcohol consumption (OR = 2.33, 95% CI [0.98–5.56]) increased the risk of LTBI but the association was no significant. However, living in the north zone (OR = 0.25, 95% CI [0.14–0.47]) or suburb zone (OR = 0.19, 95% CI [0.09–0.39]) and having a secondary education level (OR = 0.49 95% CI [0.29–0.83]) were associated with a significant reduction in LTBI risk. The model had 5.1% sensitivity and 99.0% specificity with a Cox & Snell $R$-square coefficient of 0.069 (6.9%). The bootstrapping results corroborated the findings of the logistic regression model.

## DISCUSSION

The risk factors associated with the acquisition of LTBI in the general population are rarely reported, and such studies are scarce (*Chen et al., 2015*; *Martinez et al., 2013*; *Ncayiyana et al., 2016*; *Yap et al., 2018*). This is the first study that evaluated the prevalence of LTBI, and the associated factors in the general population in Colombia, using two thresholds to classify TST positivity. A cut-off point ≥10 mm was used as the main criterion to define the positivity to *M. tuberculosis* infection following CDC guidelines (*ATS, 2000*), and it was found that LTBI prevalence (25.3%) was lower than the reported prevalence in previous studies, which also used the TST, conducted in patients from a trauma unit (38%) and health care workers (36.8%) in Cali, Colombia (*Alzate et al., 1993*; *Barbosa et al., 2014*). Compared to population-based studies where TST-10 positivity was the only diagnostic test used to identify *M. tuberculosis* infection, the LTBI prevalence observed in this study was lower than that reported in the source population of Medellín-Colombia (42.7%) and in urban informal settlements of Lima-Peru (52%) and Johannesburg-South Africa (34.3%) (*Del Corral et al., 2009*; *Martinez et al., 2013*; *Ncayiyana et al., 2016*) but higher than the estimated prevalence by mathematical models based on estimates of ARI from TST surveys and smear positive TB prevalence for the Americas region (11%) (*Houben & Dodd, 2016*) and that of other countries with a high

TB incidence such as China (20% by IGRA in adults of a rural area) and Singapore (12.7% by IGRA in urban area residents) (*Chen et al., 2015*; *Yap et al., 2018*). These observations suggest that the LTBI prevalence in urban settlements is high and variable. The differences found with respect to other studies can be partially explained by the type of population. This study excluded participants with any underlying chronic diseases that may predispose them to the infection, and most of the included participants had low and medium household SES.

Considering the possible effect of BCG vaccination on TST specificity, a higher threshold was used (≥15 mm) in individuals who had vaccinated showing a prevalence of 13.2% that coincides with the LTBI prevalence reported in Singapore (*Yap et al., 2018*). The use of a TST-15 threshold in some studies has shown a higher likelihood of detecting *M. tuberculosis* infection (*Wang et al., 2002*) and is a predictor of progression to TB comparable to IGRA (*Abubakar et al., 2018*). In the present study, it is thought that the effect of vaccination was minimal because, in Colombia, this vaccine is administered in a single dose at birth, and as has been demonstrated, its effect on the outcome of a TST decreases after 15 years (*Wang et al., 2002*), and 82% of the participants were over 30 years old. No association was found between the absence of a BCG scar and TST positivity; this observation is in line with that reported in other studies conducted in South Africa, a high TB burden setting where BCG is given at birth, and the TST is performed more than 10 years later (*Farhat et al., 2006*; *Mahomed et al., 2011*).

Using a TST-10 threshold, it was found that age, male sex, being employed, and low alcohol consumption increased the risk of LTBI, while living in the north zone or suburb zone decreased it. A large number of TB cases reported in Cali are located in the central zone including districts 9 and 11, an area that is characterized by a vulnerable population that lives in conditions of overcrowding, malnutrition, and drug use, and living in zones distant from this point reduced the LTBI risk. On the other hand, our data indicate that after 30 years, the risk of LTBI increases significantly compared with the 14–19 years age group. This is in agreement with other studies where they report a high LTBI prevalence with advanced age (*Belo & Naidoo, 2017*; *Chen et al., 2015*; *Gao et al., 2015*; *Lee et al., 2014*; *Yap et al., 2018*). It is unknown whether the increase in age is the risk factor for acquiring LTBI (*Lee et al., 2014*) or if the increase in age from 30 to 39 years reflects a cumulative exposure to people with TB within the community, allowing the development of a detectable immune response against *M. tuberculosis* infection (*Belo & Naidoo, 2017*; *Do Prado et al., 2017*; *Gao et al., 2015*).

It was found that the LTBI prevalence in males was significantly higher than that in females, especially in the 40–49 years age group. This finding matches those observed a in rural population of China and community-based studies in South Africa and Peru showing epidemiological differences in LTBI prevalence based on sex (*Chen et al., 2015*; *Martinez et al., 2013*; *Ncayiyana et al., 2016*). This is consistent with the ratio (2:1) between males and females observed in TB epidemiological studies (*Rhines, 2013*). Two possible explanations are proposed: (1) a large proportion of females remain at home and are less likely to be exposed compared to males who have more active social responsibilities (*Gao et al., 2015*; *Kizza et al., 2015*), and (2) there is a differential susceptibility to

*M. tuberculosis* infection or predisposition to delayed-type hypersensitivity responsiveness that is dependent on sex (*Verhagen et al., 2012*). Being employed was associated with an increased LTBI risk. One possible explanation is that the transmission of *M. tuberculosis* occurs in spaces where the working population is exposed, for example, public transport, which is overcrowded and has little ventilation, increasing the possibility of acquiring *M. tuberculosis* infection between people who often use this type of transport to travel long distances to their work and are repeatedly exposed (*Oni et al., 2012*). In support of these findings, sex and employment status showed a significant association ($p = 0.000$). In fact, 63% of females (mostly housewives) were unemployed, while 73% of males worked outside the home.

Our results show an increase in the risk of TST positivity in light drinkers compared to nondrinkers. Previous studies have shown that, dependent on the amount consumed, alcohol intake is associated with an impaired immune system, increasing susceptibility to respiratory infections such as pneumonia and tuberculosis (*Happel & Nelson, 2005*; *Rehm et al., 2009*; *Silva et al., 2018*). The relationship between low levels of alcohol consumption and TB risk remains unclear. *Soh et al. (2017)*, in a cohort of middle-aged and elderly Chinese adults, found that low-dose intake of alcohol (monthly to weekly frequency) was associated with a lower risk of TB compared to nondrinkers, but their observation was limited only to nonsmokers. In contrast, in current smokers the consumption of alcohol at low levels did not show any protective effect for the development of TB, but the intake of two or more drinks daily acted synergistically with smoking to increase the TB risk. Similarly, in a study of household and community contacts in India, it was observed that male sex, alcohol consumption and smoking were risk factors for TST positivity. In the multivariate analyses, these three variables also showed an association with LTBI risk (OR = 3.93, 95% CI [1.3–11.9]) (*Narasimhan et al., 2017*). These studies suggest that alcohol and smoking are strongly correlated. In the present study, a low proportion of individuals smoked (11.5%), so the influence of this factor on the findings was ruled out. The small number of participants with moderate and high alcohol consumption could make comparisons between the categories difficult, hiding any possible association with TST positivity. Given the low rate of TST positivity, the lack of reactivity due to the immunosuppressive effects of alcohol cannot be ruled out.

When the TST-15 threshold was used, results similar to those observed with TST-10 were found. Being male and having low alcohol consumption increased the LTBI risk. In contrast, residing in the north zone or suburb zone and having a secondary education level decreased risk within the model. The association between a higher level of education and a reduction in the LTBI risk can be explained by an improvement in the quality of life of individuals and awareness of health risks dependent on lifestyle habits, and thus increased efforts to reduce their exposure to recognized TB risk factors such as poverty, overcrowding, smoking, and malnutrition (*Lonnroth et al., 2009*). These findings coincide with the results found in other population-based studies in China, Singapore and South Africa (*Chen et al., 2015*; *Ncayiyana et al., 2016*; *Yap et al., 2018*).

The average induration size, using both TST-10 and TST-15 thresholds, was above the cut-off points, allowing us to rule out any possible effects of BCG vaccination or cross-reactivity with nontuberculous mycobacteria (*Borroto et al., 2011*). This suggests that

the threshold of ≥10 mm in the city is a useful tool to confirm *M. tuberculosis* infection in agreement with the recommendation of the CDC (*ATS, 2000*), while the threshold ≥15 mm can be used as an increased risk indicator for the development of TB in asymptomatic individuals (*Ministry of Health, 2010*). In support of this recommendation, a cross-sectional study in the Ethiopian population (TB incidence: 164/100,000 inhabitants) using a threshold TST-10 showed that the average TST induration size in community controls was 7.9 mm, which was less than that observed in our study (16 mm), while in household contacts and patients with TB, it was 13.6 mm and 18.1 mm, respectively (*Shero et al., 2014*).

The TST induration size in the generalized linear model was influenced by the increase in age from 40 years, a higher level of education, residing in the north zone or suburb zone, low alcohol consumption and being underweight, which coincides with the variables previously associated with TST positivity in the logistic models for TST-10 and TST-15, except for the association with BMI < 18.5 $kg/m^2$. The results found show a negative association between underweight and TST induration, explaining the low average TST induration size observed in these individuals compared with normal weight, overweight or obese individuals. These results contrast with previous studies where it has been observed that a lower BMI is an important risk factor for the development of TB (*Hanrahan et al., 2010*; *Lonnroth et al., 2010*; *Patra et al., 2014*). Additionally, overweight and obesity could decreased the risk of active TB (*Hanrahan et al., 2010*; *Kim et al., 2018*; *Lin et al., 2018*). The relationship between BMI and LTBI risk is not well described (*Chen et al., 2015*; *Saag et al., 2018*). Several population-based studies in rural areas of China have shown that overweight and obesity significantly increase the LTBI risk. These studies have also reported a nonsignificant negative association between a lower BMI and LTBI (*Chen et al., 2015*; *Gao et al., 2015*). Associations similar to those observed in our study, with crude ORs of 0.18 for TST-10 and 0.35 for TST-15, were found.

There were some limitations of this study. First, its cross-sectional nature did not allow the establishment of temporality or causality between LTBI and the associated factors. Second, individuals from some zones of the city (east and southeast) were not included, which implies selection bias, given that people living in the eastern area have a high TB risk. Third, since there is no gold standard for LTBI diagnosis, the estimation of its prevalence could be affected by TST performance. Fourth, there is a possibility of misclassification of drinkers by a self-report bias and the ability to remember. Fifth, as in any observational study, there could be a residual confounding effect of unknown or unmeasured factors on the associations observed. Despite these limitations, this study had an adequate sample size and statistical power and was the first population-based study of LTBI prevalence and associated risk factors in Colombia, so it provides valuable information in a country with an intermediate TB burden, where BCG is administered at birth.

## CONCLUSIONS

The LTBI prevalence in our population without associated comorbidities, as measured using two thresholds TST (≥10 mm and ≥15 mm), was moderate (25.3% and 13.2%, respectively), reflecting a significant TB burden and the ongoing transmission of

*M. tuberculosis* in the community. Several risk factors traditionally associated with TB (age, educational status, sex, employment situation, BMI and alcohol consumption) showed an association with the positivity and induration of TST in the three multivariate models. Unexpectedly, a lower BMI ($<18.5$ kg/m$^2$) showed a negative and significant association with TST induration, and the LTBI prevalence in underweight individuals was low. In contrast, studies have shown an increase in TB risk among underweight individuals. Additional studies are required to validate our findings and identify other risk factors associated with LTBI. Given that BCG vaccination does not confer protection against TB in adults, and most people who develop TB in Colombia and other developing countries are vaccinated. The community identification of high-risk groups and prophylactic LTBI treatment to prevent progression to TB could be a cost-effective strategy of great impact.

## ACKNOWLEDGEMENTS

We thank all participating institutions, directives and staff for their support with this study.

### Funding
This work was supported by an internal research grant from the Universidad del Valle. The funders had no role in study design, data collection and analysis, decision to publish, or preparation of the manuscript.

### Grant Disclosures
The following grant information was disclosed by the authors:
Universidad del Valle.

### Competing Interests
The authors have no conflicts of interest to declare.

### Author Contributions
- Javier Andrés Bustamante-Rengifo conceived and designed the experiments, performed the experiments, analyzed the data, prepared figures and/or tables, authored or reviewed drafts of the paper, and approved the final draft.
- Luz Ángela González-Salazar performed the experiments, authored or reviewed drafts of the paper, and approved the final draft.
- Nicole Osorio-Certuche performed the experiments, authored or reviewed drafts of the paper, and approved the final draft.
- Yesica Bejarano-Lozano analyzed the data, prepared figures and/or tables, and approved the final draft.
- José Rafael Tovar Cuevas analyzed the data, authored or reviewed drafts of the paper, and approved the final draft.
- Miryam Astudillo-Hernández conceived and designed the experiments, prepared figures and/or tables, authored or reviewed drafts of the paper, and approved the final draft.

- Maria del Pilar Crespo-Ortiz conceived and designed the experiments, prepared figures and/or tables, authored or reviewed drafts of the paper, and approved the final draft.

## Human Ethics

The following information was supplied relating to ethical approvals (i.e., approving body and any reference numbers):

This study was approved by the ethics committee of the Universidad del Valle-CIREH (#008-015).

## Data Availability

The raw data are available as Supplemental Files.

## Supplemental Information

Supplemental information for this article can be found online at http://dx.doi.org/10.7717/peerj.9429#supplemental-information.

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
