# Peer review of "Prevalence of and risk factors associated with latent tuberculosis infection in a Latin American region"

_PeerJ, doi:10.7717/peerj.9429_

## Round 0.1 · original submission · Major Revisions

As suggested by the reviewers, I agree that there is merit for publication of your manuscript. However, the manuscript needs some major revisions especially with the language (I strongly suggest getting it corrected by a professional in the English language). The reason is not merely on the grammatical errors but on the overall message conveyed with the right tone. Please also include appropriate statistical analysis. Please carefully consider the suggestions made by the reviewers and I hope it can help you improve the manuscript. I look forward to receiving a revised version.

·

Basic reporting

• The English language should be improved to ensure that an international audience can clearly understand your text. For example, the inconsistent use of abbreviations and certain terms/words.
• I have annotated and highlighted the PDF for errors, abbreviations and other sections of the manuscript which need to be corrected.

Experimental design

No comment

Validity of the findings

I thank the authors for providing the raw data. Although study results are compelling, there are few issues which need to be addressed/explained in order to improve this manuscript.

• Authors imply the study’s findings represent epidemiology and risk of LTBI in the general population, but this cannot true. Study participants were recruited from healthcare facilities (3 primary healthcare facilities/clinics and 1 hospital), Health facility-based participants do not represent people with LTBI in the general population.
• Authors have not made a compelling justification of fitting different models for different cut-offs and skin induration. I would suggest they only present a model for ≥10 mm cut-off in the main text. And other models can be presented as supplementary materials.
• Authors present findings and discuss findings on joint effects such as education and BMI. It is not clear what they mean by joint effects. They also need to describe in the analysis section how they assess joint effects

Additional comments

I commend the authors the important study on one of public health problem in Colombia. In addition, the manuscript is clearly written in professional. If there is a weakness, it is in the statistical analysis (as I have noted above) which should be improved upon before Acceptance.

·

Basic reporting

Double check figures and tables
• The figures and tables were double-checked.
• There were no clear titles for the figures and tables
• Figure 2, the age categories will read better as years instead of year for example 14-19 years not 14-19 year.
• Table 1 the average age stated in line 187 was not found in the table, BMI was not labeled correctly (Table 1 and 4).
• Table 3 the age groups start at 40-49.
• Highlighting the significant variables in the tables (bolded/ by using and asterisks * -used in text as a pointer to an annotation or footnote) would simplify interpretation of the tables.

Open the raw data
• I managed to open raw data and it was well described in English, the dataset had information on the 589 participants.

English language check
• Language is clear, intelligible and professional but some sentences could be paraphrased.

Basic Reporting
Clear and unambiguous, professional English used throughout
• The article was written in English, fairly clear, unambiguous and technically correct. The professional standards of courtesy and expression.
• For the abstract line 24 and identifying the associated risk factors reads better than knowing the associated risk factors.
• The authors started most sentences with numbers, line 29, 120, 189 and it does not read well please paraphrase and start the sentences with words.
• Please paraphrase lines 30-33 …Logistic regression showed that being aged between 40 to 69, male, employed and low alcohol consumption were risk factors for TST positivity , while living in the north and suburb zone and having secondary education were protective factors against TST positivity.
• The use of gender (men and women) throughout the manuscript could be replace with sex. Please stick to using females and males (14 year olds cannot be classified as men and women). Sex is a biological concept based on biological characteristics such as differences in genitalia and on the other hand, gender primarily deals with personal, societal and cultural perceptions of sexuality.
• Paraphrase lines 37-39 ...Being an employed male, over 40 years of age with lower level of education, low alcohol consumption and being overweight should be a priority group for the prophylactic treatment as a strategy for the TB control in the city.

Experimental design

The primary research is within the Aims and scope of the journal.
Introduction
• The introduction is well done and the reviewed literature is properly referenced.
• Maybe introduce the different cut off points for the TST in paragraph 4 (Lines 85-92).
• The aim is clear and to the point. The research question is well defined and relevant.

The investigation was rigorous and performed to a high technical and ethical standard.

The methods were described in detail.
Materials & Methods
• There is no clear justification for the selection of the 4 hospitals (line 107).
• Line 108-109 please introduce the zones fully. It is not clear whether north, suburb and center were the only zones in Cali.
• Line 151 and 551-552. MinSalud, its not clear whether it is a citation or not, there is no year.
• Line 169 is or the proper conjunction to use.

Validity of the findings

Results
• Line 186 there is typo its 14-70 not 19-70 years-old.
• Paraphrase lines 186-187.
• Line 200 its males not male.
• Lines 202-203, males, and females not male and female. Please consider exchanging (line 202) with growing age to increasing age.
• Line 232 from table 3 does not have age group 3o above it starts at 40-49 years.
• Line 241 specify that the results presented are for the bivariate analysis.
• Line 243 male should be males and female females. Plural and not singular.
• Paraphrase line 253-258…it was found that being aged 30 and above with OR from 4.29-8.30, being male [OR = 1.71, CI not IC 95 1.04-2.84], Please consider exchanging being employed not be active at work …
• Line 275, please consider exchanging belonging to the male gender to being male.

Discussion
• Lines 293-300, the sentence is too long, please restructure. Remove the citations within the sentence and place the citations at the end of the sentence.
• Paraphrase 301-304, please consider exchanging …without chronic diseases underlying to any underlying chronic diseases.
• Paraphrase lines 306-308. Please remove the citations within the sentence and place the citations at the end of the sentence. It can be …the LTBI prevalence reported in Singapore (Yap et al., 2018).
• Paraphrase lines 333-337. Please consider exchanging men with males and women with females (lines 333 and 337-338) and place the citations at the end of the sentence.
• Paraphrase lines 350-353, the sentence is too long.
• Lines 360-367 there is repetition, Narasimhan et al (210) is not correctly cited and it not in the reference list. There is a typo in line 362 it is in not In.
• Line 377… Please consider exchanging …Being male and low alcohol consumption increased the LTBI risk not the male gender.
• Lines 382-384 restructure the sentence and place the citations at the end of the sentence.
• Paraphrase lines 391-394 Shero et al (Shero et al., 2014) can be removed and the citation can be at the end of the sentence for better comprehension.
• Paraphrase lines 402-405 cite properly.
• Lines 408 and 409 the numbers (170) and (187) are misplaced.
• The zone East and South were mentioned for the first time in line 419.

Conclusion
• It is well stated and linked to the aim and limited to the stated limitations

The underlying data has been provided and are robust, statistically sound and controlled.

Additional comments

• The writing style and the English language can be improved to ensure that the paper can be clearly understood by the international audience, the current phrasing makes comprehension difficult.
• Citations can be presented better to help with the flow of ideas and to avoid repetition.
• Presentation of results should be improved upon before acceptance.

Reviewer 3 ·

Basic reporting

- Ethics Statement should be under its own section and not combined with participant information
- Global comment: grammar and writing style not very strong throughout and needs to be reworked. For e.g. abstract, lines 193-194 should be in past tense, etc.
- Common writing guidelines should be followed. For e.g. M. tuberculosis can be abbreviated to Mtb, after first mention
- In line 184 the title reads “Socio-demographic and behavioral characteristics”, but the authors don’t discussion any behavioral traits. I would assume that mentioning tendencies of mood, depression, anxiety, etc. would constitute behavioral traits and not just addiction patterns.
- Line 272/73 are repetitive
- Multiple times in the results, authors use the term ‘low alcohol consumption’. This is very misleading in the style it is currently written and only in line 350 does the comparison become clear. Needs rewording
- Line 407: are overweight and obesity really protective factors? Needs rewording

Experimental design

- Line 274 and 254-257: being an employee or being active at work leads to increased LTBI risk? Poorly worded
- Lines 260 and 280: very poor sensitivity

Validity of the findings

- Lines 294-305: authors are comparing LTBI prevalence in this study to other population-based studies, but are the evaluation criteria the same? There’s no mention of what criteria these studies used, especially was it TST alone or TST+IGRA? Doesn’t seem to be a fair comparison
- Lines 344-347: authors claim that employment (being employed) is a risk factor for LTBI. Although they provide some reasoning as to why this may be, this is still a lofty claim. For this to be substantiated, there needs to be additional analysis done to correlate the ‘type’ of employment, for e.g. the nature of work seen in subjects from low-income strata vs those individuals that work in corporate/office jobs
- The authors at multiple occasions suggest ‘lack of gold standard’, but globally Quantiferon is still considered highly indicative of LTBI status

Additional comments

Significant drawbacks of this manuscript include lack of IGRA testing among subjects and/or comparative analysis with similar studies from other parts of the world, which have used TST 10/15mm for detecting LTBI prevalence

---

## Round 0.2 · accepted · Accept

Thank you for considering the productive comments by the reviewers and improving the manuscript. I am happy to accept reviewers decision and recommend you manuscript for publication. Congratulations.

·

Basic reporting

No comment

Experimental design

No comment

Validity of the findings

No comment

Additional comments

The authors made the suggested corrections to my satisfaction. The manuscript reads well now.

Reviewer 3 ·

Basic reporting

Reads well, all previous concerns addressed

Experimental design

Study design is fine

Validity of the findings

Relevant findings reported

Additional comments

Manuscript can be accepted for publication